# Two Semiconductor Companies’ Financial Support Compensation (FSC) Programs for Semiconductor Workers with Suspected Work-Related Diseases (WRDs)

**DOI:** 10.3390/ijerph19148694

**Published:** 2022-07-17

**Authors:** Hyoung-Ryoul Kim, Dong-Uk Park

**Affiliations:** 1Department of Occupational and Environmental Medicine, College of Medicine, The Catholic University of Korea, Seoul 06591, Korea; cyclor@catholic.ac.kr; 2Department of Environmental Health, Korea National Open University, Seoul 110-791, Korea

**Keywords:** semiconductor operation, work-related diseases (WRDs), the financial support compensation (FSC) for WRDs

## Abstract

This study described two companies’ financial compensation programs for semiconductor workers with suspected work-related diseases (WRDs) and discussed the major related issues. The key contents of the programs found on the websites opened by two semiconductor companies (Samsung and SK Hynix) were cited. In order to select the suspected WRDs for the FSC, all available epidemiologic studies related to health problems conducted in the semiconductor industry were reviewed. Most program contents are similar, although the amount of financial compensation and a few types of disease available for compensation differ between the companies. The group of cancer, rare disease, childhood rare disease among children born to semiconductor workers (hereafter selected diseases among offspring), and fetal loss, including spontaneous abortion (SAB) and stillbirth, were considered for compensation. An employment duration of longer than one year on a semiconductor production line is required for FSC for cancer or rare disease. The FSC for SAB and offspring disease require a period of employment longer than one month, either before three months prior to conception or between conception and childbirth. The maximum amount of compensation per type of cancer and rare disease was fixed based on the medical treatment fee. The FSC programs of the two companies have been operated successfully to date. These programs are arguably considered to contribute to resolving the conflict between companies and workers with WRDs.

## 1. Introduction

Workers in semiconductor wafer fabrication and packaging in the semiconductor industry have the potential to be exposed to a broad number of chemicals involved in the related manufacturing processes, which have been changing dramatically over the years with the rapidly advancing technology applied in this industry. Potentially hazardous chemicals such as metals, photoactive chemicals, organic solvents, acids, and toxic gases are used in the semiconductor industry in a wide variety of combinations and workplace settings, resulting in additional by-products. Over the past decade, there has been speculation that semiconductor operations and the related working environments and jobs, especially before the 2000s, may be related to the risk of various types of cancer, rare diseases, and reproductive toxicity. However, this has not been clearly examined [1].

Controversy over health risks among semiconductor workers began growing in 2007 when a female semiconductor worker aged 22 died from leukemia. Her case was initially denied by the Korea Workers’ Compensation and Welfare Service (KWCWS), but it was later awarded compensation as an occupational disease by the administrative court. Since then, a number of former semiconductor workers and groups representing the alleged victims have continued to claim for over a decade that their chronic diseases, including certain types of cancer and rare diseases, were caused by semiconductor operations and jobs.

Various actions have been taken to resolve the social dispute originating from the chronic health risks, including cancer, faced by semiconductor workers. Several studies have been undertaken to examine occupational factors that may be associated with health risks among semiconductor workers [2,3,4]. No companies in Korea have an incentive to willingly compensate or support workers with work-related diseases using their own compensation programs given that most companies are already registered under the Industrial Accident Compensation Insurance Act (IACIA) that has been operated as a kind of social welfare program. In addition, companies have no legal responsibility to internally compensate workers with a WRD irrespective of the national compensation program. 

From 2016 to the present, two semiconductor companies and one display manufacturing company in the Republic of Korea (Korea) have developed financial support compensation (hereafter referred to as FSC) programs to provide financial support to workers suffering from cancer, rare diseases, reproductive diseases, and selected diseases among their offspring, if minimal job profile requirements are satisfied. They have operated similar FSC programs that vary in detail, such as the year started, amounts of financial support, and types of disease compensated. This study aimed to describe the two semiconductor companies’ FSC programs for semiconductor workers and to discuss major issues, including the background, implications, and scientific basis for the FSC programs. In order to provide a scientific basis for the FSC program, we cited the key results of epidemiological studies conducted on the semiconductor industry examining the risk of cancer, rare diseases, and reproductive health problems.

## 2. Materials and Methods

### 2.1. Brief Description of the Semiconductor Manufacturing Process and Major Health Hazards

A semiconductor company manufactures both wafer fab and chip packaging. The fab for integrated circuits or chips is processed separately in the so-called fab clean rooms (hereafter referred to as a production line). Integrated circuits are fabricated onto a silicon wafer through a series of repetitive processes composed of four main operation groups. The principle of semiconductor operations and the major health hazards, including various classes of chemicals and electromagnetic fields generated in these operations, has been comprehensively described elsewhere [5,6,7].

### 2.2. Selection of Suspected Work-Related Diseases (WRDs) for the Financial Support Compensation (FSC)

This study introduced the FSC programs operated by two semiconductor companies (Samsung and SK Hynix) that provide financial support to semiconductor workers who have suffered WRDs such as cancers, rare diseases, and reproductive health problems (hereafter referred to as a suspected WRD). These programs are independent of the national compensation program operated by the Korea Worker’s Compensation and Welfare Service under the IACIA. The key contents of the programs available on websites provided by the two companies for the registration are summarized and described [8,9].

Suspected WRDs for the FSC were selected based on the examination of the following review process or other sources. Firstly, types of cancer, reproductive health problems, and rare diseases for which increased risks have been found in the semiconductor industry were included in the group of diseases for the FSC. All epidemiologic studies conducted in silicon wafer fab operations in the semiconductor industry were collected through an extensive literature review of articles reported from 1988 through the end of 2015 without limitation on the study period. The keywords used in the literature search were “semiconductor operations”, “wafer fab operation”, and “clean room environment”, singly and in combination. All types of cancer, rare disease, and reproductive toxicity that were evaluated at least once as significant were selected for the FSC. Secondly, types of cancer and rare diseases already compensated by either KWCWS or the administrative court were also included in the FSC. A few types of cancer and rare diseases examined in epidemiological studies or investigations conducted in their plants were included for the FSC programs, which differ slightly between companies.

### 2.3. Job and Disease Factors for Calculating the Amount of Financial Support Compensation (FSC)

A formula combining job and disease factors for calculating the amount of the FSC was developed by the type of diseases compensated. The maximum amount of the FSC per type of diseases was first fixed based on the associated medical treatment fee. Job factors that influence the development of suspected WRDs in semiconductor production lines and also disease factors related to the burden of treatment expense were also identified and combined to calculate the FSC amount. A contribution fraction by job and disease factors was developed and applied to the total FSC amount. The formulas combining disease and job factors were created in order to avoid any prejudice through quantitative evaluation on the contribution to the development of a suspected WRD.

### 2.4. The Operation of the FSC Programs

A Financial Support Compensation Committee (FSCC) has been created by each company comprised of fewer than ten external occupational health professionals and lawyers. The number by type of professional and operation methods differs between companies. They have been operated entirely independent of the companies as well as the KWCWS. The key roles of the committees are to evaluate whether applicants are eligible for the FSC programs, calculate the FSC amount based on the designated formula combining several job and diseases factors, and determine the FSC amount to each company and applicants. As soon as a company receives the FSC results from the FSCC, it sends the amount of the FSC directly to the applicant (Figure 1).

## 3. Results

### 3.1. Overview of the FSC Programs

Most of the components of the programs, including the type of the suspected WRD, requirements for registration, and principles and methods for the FSC, are nearly the same (Table 1). However, the amount of the FSC by disease differs and minor changes in the types of disease available for the FSC might be made during operation, which are not currently reported. All incumbent or former workers in the companies and their subcontractors involved in the semiconductor production line are eligible to apply for the FSC program, regardless of the type of operation in which they worked or the job they performed. Only contractors working in production lines within the company are subject to the FSC. Both external contractors outside semiconductor plants and contractors involved in supply and maintenance tasks for various process equipment and machines were excluded because they are also exposed to hazardous agents generated from other companies. The range of semiconductor workers covered was decided by an agreement between the company and a labor union or a group representing victims in the case of no labor union. The FSC program is carried out by a third party consisting of external professionals. The deadline for applying for the FSC with Samsung is set as the end of 2028, although it could be extended depending on the circumstances. Semiconductor workers who either filed a lawsuit or intend to file against either company are not eligible for the program, while victims who received the FSC are still allowed to register with the KWCWS.

### 3.2. Selection of the Suspected WRD for the Financial Support Compensation (FSC)

Certain types of cancers, reproductive health problems, and rare diseases were included as the suspected WRDs candidates for the FSC. The types of cancer, rare diseases, and reproductive health problems that have been found to be significant in epidemiological studies published in international journals were first selected for the FSC, regardless of the consistency among the literature or the type of studies (Table 2) [3,10,11,12,13]. A total of 11 cancers deemed to be significant (NHL (no. of studies = 1), brain (n = 1), lung (n = 1), thyroid (n = 1), ovarian (n = 1), stomach (n = 1), melanoma (n = 2), rectal (n = 2), prostate (n = 2), pancreatic (n = 2), and breast (n = 2)) were included for the list of FSC. No epidemiological study examining the risk of rare diseases other than cancer has been conducted for semiconductor workers.

The types of reproductive health problems evaluated to be significant in the semiconductor industry were spontaneous abortion (SAB) [14,15,16,17,18,19,20], prolonged time to pregnancy [21], and congenital anomalies in offspring of male semiconductor workers [22]. In particular, Lin et al. [22] found an increased risk of death from congenital anomalies (adjusted OR, 3.26; and 95% confidence interval [CI], 1.12–9.44) and heart anomalies (adjusted OR, 4.15; 95% CI, 1.08–15.95) in the offspring of male workers (n = 5702 children) who were employed during the two months prior to conception. Among several reproductive health problems, congenital anomalies, fetal loss, and SAB with medical certification were included as subject to the FSC [22]. Based on the studies reporting an increased risk of SAB and congenital anomalies among semiconductor workers, fetal loss, including spontaneous abortion (SAB) and stillbirth, and selected diseases among the offspring of semiconductor workers, such as congenital, childhood cancer, and rare diseases (hereafter selected diseases among offspring), were eventually included in the FSC list (Table 3).

Leukemia, brain cancer, ovarian cancer, and multiple sclerosis were rejected as occupational diseases by the KWCWS but were later awarded compensation by the administrative court (Table 2), even though they have not been examined in epidemiological studies. There have been a few types of cancer and rare disease included for the FSC by each company through internal investigation, epidemiological study, or compromise, but they slightly vary between the two semiconductor companies (Table 3). Detailed information on the suspected WRDs each company has included in the FSC list has not been reported, even though most of the suspected WRDs for the FSC are the same.

### 3.3. The Requirements for Semiconductor Workers to Register with the Program

The requirements for which semiconductor workers can register with the FSC program were decided by the type of the suspected WRD (Table 4). The minimal job profiles that are required by the FSC program to prove a work association with the disease are far less strict than those required by the KWCWS. An employment duration longer than one year on a semiconductor production line is required for FSC for cancer and rare disease. FSC for fetal loss, including SAB, stillbirth, and selected diseases among offspring, requires a period of employment longer than one month, either before three months prior to conception or between conception and childbirth. These minimal job requirements, including employment duration, are designed to expand the number of semiconductor workers for the FSC.

### 3.4. Job and Disease Factors for Calculating the Amount of the Financial Support Compensation (FSC)

The amount of the FSC was designed to be smaller than that provided by the KWCWS but greater than the required medical treatment for the suspected WRD involved. The maximum amount of the FSC per type of cancer and rare disease was first fixed based on medical treatment fees by disease (Table 5). Several characteristics related to job and disease were categorized according to the relative contribution to the development of cancer, rare diseases, and offspring disease and applied to the formula for calculating the FSC amount. Details related to the calculation of the FSC amount, such as the formula combining job and disease factors and the contribution rate to the development of the suspected WRD and its components, have not been reported. The formula combining disease and job factors was created in order to avoid any prejudice through quantitative evaluation on the contribution to the development of a suspected WRD. The same FSC amount was determined for fetal loss, including SAB and stillbirth. The specific amounts of the FSC differ between companies and have not been reported. Official documents related to the calculation basis for the FSC amounts, the amount compensated, and the statistics regarding the FSC have not been reported beyond summary results reported in the media [8,9]. 

## 4. Discussion

This study described the FSC programs that are being supported by two semiconductor companies for semiconductor workers with cancer, rare diseases, reproductive health problems, and selected diseases among their offspring. All countries have compensation-based surveillance systems and/or non-compensation-based surveillance systems, or both, to recognize occupational disease under a social welfare act or other pertinent regulations [23]. As far as we reviewed, there has been no peer-reviewed literature reporting on the FSC programs that companies have developed to financially support workers with a suspected WRD, independent of national compensation programs.

It is well-known that the semiconductor industry involves rapidly shifting technologies and drastic changes in chemical use, processes, and technology, multiple exposures, and more, making it difficult to examine the hazardous agents or jobs causing chronic health risks. Most semiconductor workers suffering cancer and rare diseases in Korea have been denied national compensation because it is difficult for them to prove the association between various types of chronic disease and past exposure from a rapidly changing work environment and process.

Before the FSC programs were introduced by the two semiconductor companies (Table 1), former semiconductor workers with a suspected WRD and civic groups had staged a time-consuming legal battle to prove that their diseases were closely linked to a harmful working environment. A suit was filed with the administrative court after the KWCWS rejected their compensation requests. Among a total of 55 semiconductor workers registered with the KWCWS up to the end of 2017, only 19% (n = 10) were awarded compensation from the KWCWS [7], indicating that it is very difficult for semiconductor workers to be entitled compensation through this avenue. Most of the semiconductor workers who registered with the KWCWS were found to be employed at the two largest semiconductor companies. Since 2016, these two semiconductor companies have operated the FSC programs to willingly provide their workers with financial compensation, including medical treatment fees, if minimal job profile requirements were satisfied (Table 1 and Table 4). The semiconductor company’s FSC program experiences as described below could be helpful for large companies, including high-tech operations, who are considering an internal financial assistance program for their workers who suffer from a WRD, independent of the national compensation program.

Firstly, these FSC programs are considered to have contributed to resolving a social dispute that lasted for more than a decade. Since the operation of these programs, no social conflict related to the causality of the semiconductor environment in terms of WRDs has been reported. All semiconductor workers who receive FSC from the companies are still allowed to apply to the KWCWS.

Secondly, semiconductor workers with several rare diseases, including cancer and selected diseases among their offspring, can be financially assisted by the companies if certain minimum-period job history on a production line can be demonstrated. Most types of rare diseases and reproductive and diseases among offspring are unlikely to be awarded compensation by the national program in many countries, even though they are work-related to some extent. Korea has listed 12 types of WRDs allowed compensation under the Industrial Accident Compensation Insurance Act (IACI) [24]. Occupational disease surveillance in Korea entirely relies on data provided by workers’ compensation systems, which set definitions for disease compensation. For cancer and rare diseases, significant work-related causality must be proven through epidemiological investigation in order to be entitled to national compensation. Even reproductive health problems and diseases among offspring are not listed as WRDs. Most rare diseases being compensated by the semiconductor companies (Table 3) may be likely to be rejected by the KWCWS. Other activities of life cannot be assessed or are not considered for this FSC program.

Thirdly, the two semiconductor company’s FSC programs have impacted the process of evaluating the causality of a job or environment with cancer and rare disease as required by the KWCWS. Since 2018, the KWCWS has decided to not require further epidemiological investigation for semiconductor workers who have eight types of disease, leukemia, aplastic anemia, ovarian cancer, NHL, breast cancer, lung cancer, and multiple sclerosis, because they were either compensated by the KWCWS or the administrative court as an occupational disease (Table 2). The compensation of these diseases registered with the KWCWS is now simply decided based on basic job history, including the latency period and job history in the semiconductor production line, without further epidemiologic investigation. In most industries, further detailed epidemiologic investigations of cancer and rare disease are required to be performed by a designated national institute [7]. This national compensation decision for cancer and rare diseases not only takes a long time but relies on a strictly scientific basis. Unfortunately, this approach applied to semiconductor workers by the KWCWS has not yet been expanded to other industries. Ultimately, national measures to improve the KWCWS program should be enacted so that the compensation for WRDs can be decided swiftly based on the principle of the social welfare insurance act.

Fourthly, it is meaningful that contractors involved inside the fab and chip production line inside the companies are compensated even though they are not directly employed by the parent company (Table 1). There has been a growing concern over the inclusion of contractors involved in supply and maintenance tasks for semiconductor process equipment and machines within the parent company. They are currently excluded because they are simultaneously exposed to work environments at multiple semiconductor companies. Korea does not maintain a social welfare or social insurance system that workers can typically use. The national KWCWS compensation program is the only social welfare system for facilitating the rehabilitation of workers with work-related injuries and diseases and promoting their return to society [24]. In particular, the national compensation procedures require workers with cancer or rare diseases to prove rigorous scientific causality. The Korean National Health Insurance program does not include sick leave for workers. Further collaboration is necessary to financially assist them through an FSC consortium among the two semiconductor companies.

Finally, the semiconductor companies can use the results from the FSC programs not only to identify new/emerging work-related health issues, but also to provide a signal that will initiate interventions and prevention efforts. These FSC programs should have a strong link with workplace prevention efforts, such as direct workplace interventions aimed at protecting co-workers or removing workplace risk factors and different forms of primary and secondary prevention.

This study faced clear limitations due to the lack of data on both the details of the FSC results and the formula applied to calculate the individual FSC amount. We have accessed some of the results released at different times by the two companies through the media. Samsung’s FSCC reported that its program had compensated a total of 458 semiconductor workers, including contractors, (n = 26, 6%), as of the end of May 2020, about two years after Samsung started the program in January 2018 (Table 1). A total of 14.2 billion KRW had been provided to them. Fifty-eight workers were rejected because they were ineligible for the program [9]. The number of semiconductor workers SK Hynix compensated from the initiation of the program in November 2016 up to November 2017 was 820, with a total FSC amount of 7 billion KRW [8]. Unfortunately, there have been no detailed FSC statistics classified by type of disease, type of workers, specific job profiles, etc. The FSC results have not been regularly or officially reported. In addition, no study has evaluated the responses from the semiconductor workers who received this FSC. To our knowledge, no one at all has filed a lawsuit against the companies since the programs were introduced.

## 5. Conclusions

The FSC programs supported by the two semiconductor companies have been operated successfully to date. The group of cancer, rare disease, offspring rare disease among children born to semiconductor workers, and fetal loss, including SAB and stillbirth, were considered for FSC. Semiconductor workers who suffered from a suspected WRD can be financially supported and compensated if they satisfied the minimal job-associated requirements and type of diseases. To date, no social conflict related to the causality of the semiconductor environment in terms of WRDs has been reported. These programs have contributed to resolving the conflict between the companies and semiconductor workers with the suspected WRDs. Large companies need to consider the experiences of the two semiconductor companies’ FSC programs not only to resolve similar conflicts with workers suffering from WRD, but also to provide workers with a welfare program within a company.

## Figures and Tables

**Figure 1 ijerph-19-08694-f001:**
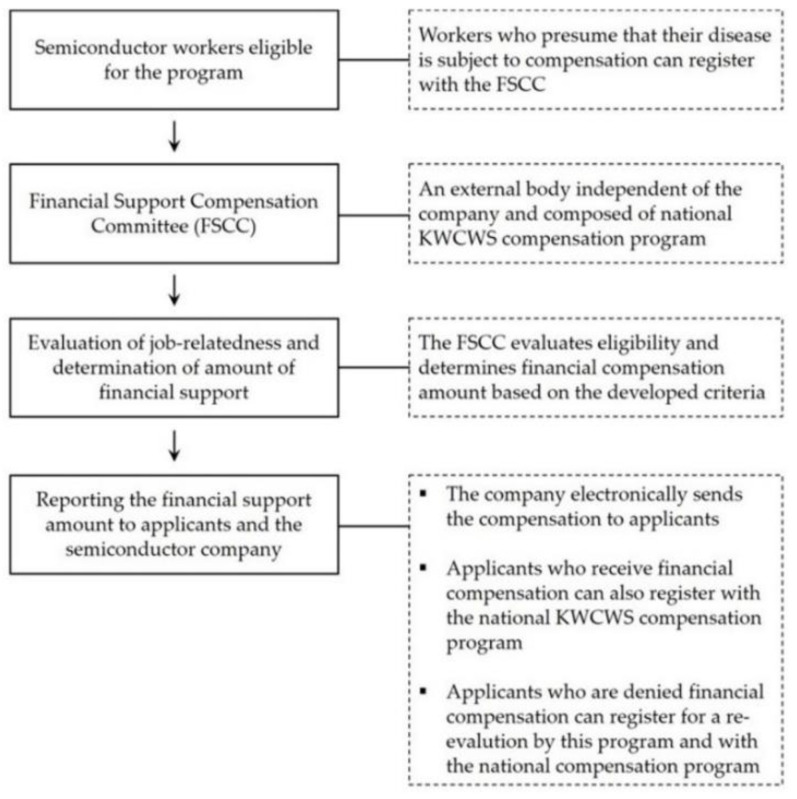
Procedures for the financial support compensation (FSC) programs for semiconductor workers with suspected work-related diseases (WRDs) operated by two semiconductor companies.

**Table 1 ijerph-19-08694-t001:** Overview of the financial support compensation (FSC) programs operated by the two semiconductor companies.

Key Components	Samsung	SK Hynix
Year initiated	January 2018 (2015 ^1^)	November 2016
Year scheduled to end	2028	Not scheduled
Inclusion of former workers	Yes	Yes
Inclusion of contractors who have worked in production lines	Yes	Yes
Registration with national compensation program	Possible	Possible
Civil or criminal suit against company during or after FSC	Not allowed	Not allowed
Operation of a Financial Support Compensation Committee (FSCC) independent of the company and the national compensation program ^2^	Yes	Yes

^1^ A financial support committee operated by the company. ^2^ The committee operating the FSC program, consisting of occupational health professionals and lawyers.

**Table 2 ijerph-19-08694-t002:** Type of cancer and rare disease either examined by epidemiological studies or compensated by KWCWS.

Disease Group	≤1 Lower 95% Confidence Interval from Cancer Risk Epidemiological Study	Compensated by Either National Compensation through KWCWS or the Administrative Court as of July 2017 [1]
Type of cancer		
Leukemia	0	Yes
Non-Hodgkin’s lymphoma	1 ^1^	Yes
Brain	1	Yes
Lung	1	Yes
Melanoma	2	No cases
Ovarian	1	Yes
Rectal	2	No cases
Prostate	2	No cases
Pancreatic	2	No cases
Breast	2	Yes
Thyroid	1 ^1^	No cases
Stomach	1	No cases
Rare disease		
Multiple sclerosis	No study	Yes

^1^ List of types of cancers with significant findings from nine cancer risk studies conducted on semiconductor operations. Number in table indicates the number of articles (out of the total of nine) showing a significant association.

**Table 3 ijerph-19-08694-t003:** Comparison of major types of cancer, rare diseases, and diseases among offspring financially supported and compensated by the two semiconductor companies as of the end of May 2022.

	Samsung	SK Hynix
Type of Cancer ^1^		
Leukemia	Yes	Yes
Non-Hodgkin’s lymphoma	Yes	Yes
Multiple myeloma	Yes	Yes
Brain	Including cancer in CNS	Yes
Lung	Including all cancer in respiratory tract	Yes
Melanoma	Yes	Yes
Ovarian	Yes	Yes
Rectal	Yes	Yes
Prostate	Yes	Yes
Pancreatic	Yes	Yes
Breast	Yes	Yes
Thyroid	Yes	Yes
Stomach	Yes	Yes
Rare cancer	22 types specified ^2^	Cancers of which the incidence is lower than 1/million
Rare disease		
Multiple sclerosis	Yes	Yes
Sjogren’s syndrome	Yes	Yes
Systemic sclerosis	Yes	Yes
Wegener’s granulomatosis	Yes	Yes
Systemic lupus erythematosus	Yes	Yes
Still’s disease	Yes	No
Idiopathic thrombocytopenic purpura	Yes	No
Idiopathic pulmonary fibrosis	Yes	Yes
Amyotrophic lateral sclerosis	Yes	Yes
Multiple sclerosis	Yes	Yes
Parkinson’s disease	Yes	Yes
Selected diseases among offspring		
Congenital	Yes	Yes
Childhood cancer	Yes	Yes
Childhood rare disease ^2^	Yes	Yes
Fetal loss including spontaneous abortion (SAB) and stillbirth	Yes	Yes

***Abbreviation***: CNS, Central Nervous System. ^1^ As of the end of February 2020, all types of cancers are included at SK Hynix. ^2^ Including all diseases that are evaluated as rare diseases by the Ministry of Health and Welfare.

**Table 4 ijerph-19-08694-t004:** The requirements for semiconductor workers who may register with the financial support compensation (FSC) program.

Major Criteria	The Requirement for Registration ^1^
Coverage of workers compensated	Former and current workers
	Former and current workers employed by contractors on a production fab line inside the company ^2^
Types of diseases compensated	
Cancer	≥ one year employment with work on or entering into a production fab line ^3^
Rare disease	≥ one year employment with work on or entering into a production fab line
Fetal loss, including spontaneous abortion (SAB) and stillbirth	Female: ≥ one month employment on a production fab line either before three months prior to conception or between conception and childbirth
Selected diseases among offspring	Male and female employees with ≥ one month employment on or entering into a production fab line either before three months prior to conception or between conception and childbirth

^1^ Most of the requirements applied to the program are the same for the two companies. ^2^ SAB and selected diseases among offspring were included for the FSC to office workers since February 2020 at SK Hynix. ^3^ Exclusion of workers who have maintained or repaired machines and equipment in production lines.

**Table 5 ijerph-19-08694-t005:** Major occupational and disease-related factors to calculate total financial support compensation (FSC).

Major Job and Disease Factors ^1^	Cancer	Rare Diseases	Selected Diseases Among Offspring	Reproductive Health Problems ^2^
Job factors				
Duration and percentage employed in this company	Yes	Yes	Yes	N/A
Level of entry into production fab line clean room	Yes	Yes	Yes	N/A
Year employed	Yes	No	No	N/A
Duration after retirement	Yes	Yes	Yes	N/A
Presence of shift work	Yes	Yes	Yes	N/A
Disease-related factors				
Age when first diagnosed	Yes			
Type of disease	Yes	Yes	Yes	N/A
Level of severity	Yes	Yes	Yes	N/A
Stage	Yes	N/A	N/A	N/A
Death	Yes	Yes	Yes	N/A
Recurrence	Yes	N/A	N/A	N/A
Fixed amount by case	N/A	N/A	N/A	Yes

^1^ The FSC amount is calculated by a formula combining job and disease factors with different weights. ^2^ Fetal loss, including spontaneous abortion (SAB) and stillbirth. Fixed amount for SAB and fetal loss.

## Data Availability

Not applicable.

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
