# Peer review of "Two Semiconductor Companies’ Financial Support Compensation (FSC) Programs for Semiconductor Workers with Suspected Work-Related Diseases (WRDs)"

_ijerph, 2022, doi:10.3390/ijerph19148694_

Round 1

Reviewer 1 Report

"The paper presents the compensation programs for semiconductor workers with work-related diseases (WRD) of two leading companies in South Korea. 
Despite the fact that authors claim that these programs are considered to contribute to resolving the conflict between companies and workers with WRD, it is questionable if such programs are equitable and efficacious at the international level. In order to be more convincing, the author should firstly present a literature review (which is almost inexistent). By doing this, authors could also underline the added value of their paper compared with other published materials.    The authors did not clearly present the methodology of their research.   The conclusions are very brief and they do not highlight the presented arguments of the results. Therefore, the conclusions must be developed.   The reference list must be developed according to the sources that the authors will investigate in the literature review part.   The paper needs proof-reading."

Author Response

Reviewer 1

Thank you very much for your highly useful suggestions, which we believe have allowed us to improve our work. We have modified and integrated the original paper taking into consideration all the comments you made. Below you can find our response to each comment/request. An English revision was made by a native English speaker.

1

"The paper presents the compensation programs for semiconductor workers with work-related diseases (WRD) of two leading companies in South Korea. Despite the fact that authors claim that these programs are considered to contribute to resolving the conflict between companies and workers with WRD, it is questionable if such programs are equitable and efficacious at the international level. In order to be more convincing, the author should firstly present a literature review (which is almost inexistent). By doing this, authors could also underline the added value of their paper compared with other published materials.    

Reply

As far as we reviewed, there has been no peer-reviewed literature reporting on the compensation programs that companies have developed to financially support workers with WRD. No companies in South Korea (and perhaps even anywhere in the world) have an incentive to willingly compensate or support workers with WRD using their own compensation programs given that most companies are already registered under the Industrial Accident Compensation Insurance Act that has been operated as a kind of social welfare program. In addition, companies have no legal responsibility to internally compensate workers with WRD irrespective of the national compensation program. We are not sure that these two semiconductor companies’ internal compensation programs can be directly applied to other companies with similar conflicts.

In accordance with your comments, we added the contents in the discussion part as follows (please see lines 238-247 on page 10)

As far as we reviewed, there has been no peer-reviewed literature reporting on the compensation programs that companies have developed to financially support workers with WRD. No companies in South Korea have an incentive to willingly compensate or support workers with WRD using their own compensation programs given that most companies are already registered under the Industrial Accident Compensation Insurance Act that has been operated as a kind of social welfare program. In addition, companies have no legal responsibility to internally compensate workers with WRD irrespective of the national compensation program.

2

The authors did not clearly present the methodology of their research.   The conclusions are very brief and they do not highlight the presented arguments of the results. Therefore, the conclusions must be developed.   The reference list must be developed according to the sources that the authors will investigate in the literature review part.   The paper needs proof-reading."

Reply

Please see our responses above. We sought related literature but were unable to find any. To clarify, we described in detail the process by which the two companies' compensation schemes were created in the methods part and the implications of these compensation systems and the possibility of social expansion in the conclusion part. Please let us know of any literature related to our topic of which you may be aware.

In accordance with your comments, the methods and conclusion have been newly added to and highlighted. Please see the methods, limitations in the discussion (please see lines 238-247) and conclusions sections.

Reviewer 2 Report

Summary: The author reviewed the compensation programs of two leading semiconductor companies in South Korea and claimed to have achieved overall success in resolving the prolonged conflict between the firm and workers (including former workers and contractors).

Comment: Can you inspire your research and assert your readers that such programs are equitable and efficacious at the international level by looking into how Intel and TSMC operate in compensating their enfeebled workers?

Without seeing the formula for compensation determination, I am hesitant to claim the program as successful. Please provide the formula (rather than just "arbitrary"), and justify that workers are fairly compensated without any prejudice. Otherwise, the postulation is far from being scientific and sound.

Finally, are the companies to decide the contribution factors, how can they be assured that the workers do not suffer from other part-time jobs or voluntary/community services that also contribute to those diseases?

Please refer to the attached PDF for minor comments on grammar and typos.

Author Response

Thank you very much for your highly useful suggestions, which we believe have allowed us to improve our work. We have modified and integrated the original paper taking into consideration all the comments you made. Below you can find our response to each comment/request. An English revision was made by a native English speaker.

Reviewer 2

The author reviewed the compensation programs of two leading semiconductor companies in South Korea and claimed to have achieved overall success in resolving the prolonged conflict between the firm and workers (including former workers and contractors).

Comment: Can you inspire your research and assert your readers that such programs are equitable and efficacious at the international level by looking into how Intel and TSMC operate in compensating their enfeebled workers?

Reply

We recommend that the experiences of the two semiconductor company’s compensation programs can be considered for large companies to resolve similar conflicts with workers suffering from WRD, but also to provide workers with welfare programs within a company.  

In accordance with your comments, we modified related sentences in conclusion part as follows (please see the lines 354-356 on page 13)

Large companies need to consider the experiences of the two semiconductor companies’ compensation programs to resolve similar conflicts with workers suffering from WRD, but also to provide workers with welfare programs within a company

2

Without seeing the formula for compensation determination, I am hesitant to claim the program as successful. Please provide the formula (rather than just "arbitrary"), and justify that workers are fairly compensated without any prejudice. Otherwise, the postulation is far from being scientific and sound.

Reply

Unfortunately, the formula has not been publicly reported to date. The companies are reluctant to elaborate the information regarding the compensation programs, including by providing the formula used to calculate individual compensation amounts and the results of compensation to date. We believe that the key variables included in the formula are sufficiently explained in the manuscript submitted. The formula combining disease and job factors were created in order to avoid any prejudice through quantitative evaluation on contribution to the development of WRD    

To clarify on the lack of compensation formulas:

The formulas combining disease and job factors were created in order to avoid any prejudice through quantitative evaluation on the contribution to the development of WRD (please see line 104-106 0n page 3]

This study faced clear limitations due to the lack of data on both the details of compensation results and the formulas applied to calculate individual compensation amounts (please see line 329 on page 12).

3

Finally, are the companies to decide the contribution factors, how can they be assured that the workers do not suffer from other part-time jobs or voluntary/community services that also contribute to those diseases?

Please refer to the attached PDF for minor comments on grammar and typos.

Reply

Individual background exposures other than jobs and environments in semiconductor operations were not assessed. Therefore, the contribution to the development of WRD by individual inheritance and background exposure due to part-time jobs or other life activities cannot be assessed or do not need to be considered for this compensation program. We have accepted all the recommendations you made in the PDF file and revised the manuscript accordingly.

In accordance with your comments, we added this point in discussion part as follows (please see the lines 289-292 on page 11)

Individual background exposures other than jobs and environments in semiconductor operations were not assessed. Therefore, the contribution to the development of WRD by individual inheritance and background exposure due to part-time jobs or other life activities cannot be assessed or do not need to be considered for this compensation program

Reviewer 3 Report

This study is a well-written thesis on health and safety policy. The method and status of economic compensation chosen by the two workplaces for the diseases of workers at semiconductor workplaces are described neutrally and well. However, it would be better if the discussion would explain the social welfare system that Korean workers can have, social insurance that does not provide sickness and sickness benefits other than occupational sickness compensation, and workers' inevitable dependence on industrial accident compensation. This explanation will show the reason why the compensation procedure for occupational diseases in Korea's industrial accident insurance has no choice but to change to a decision process based on probability rather than rigorous proof of scientific causality based on the improvement of workers' welfare.

Author Response

Thank you very much for your highly useful suggestions, which we believe have allowed us to improve our work. We have modified and integrated the original paper taking into consideration all the comments you made. Below you can find our response to each comment/request. An English revision was made by a native English speaker.

This study is a well-written thesis on health and safety policy. The method and status of economic compensation chosen by the two workplaces for the diseases of workers at semiconductor workplaces are described neutrally and well. However, it would be better if the discussion would explain the social welfare system that Korean workers can have, social insurance that does not provide sickness and sickness benefits other than occupational sickness compensation, and workers' inevitable dependence on industrial accident compensation. This explanation will show the reason why the compensation procedure for occupational diseases in Korea's industrial accident insurance has no choice but to change to a decision process based on probability rather than rigorous proof of scientific causality based on the improvement of workers' welfare.

Reply:

In accordance with your comments, we added the contents regarding the background of the social welfare system for South Korean workers as follows (please see 315-319 on page 12)

Korea does not maintain social welfare or social insurance system that workers can typically use. The national KWCWS compensation program is the only social welfare system for facilitating the rehabilitation of workers with work-related injuries and diseases and promoting their return to society [24]. The Korean National Health Insurance program does not include sick leave benefits for workers. In particular, the national compensation procedures require workers with cancer or rare diseases to prove rigorous scientific causality.

Round 2

Reviewer 1 Report

The authors made only small changes compared to the previous version of the paper. However, they explained their decision. 

Author Response

Reviewer 1

The authors made only small changes compared to the previous version of the paper. However, they explained their decision. 

Reply

We made the utmost effort to revise our manuscript in accordance with the reviewer’s comments, including yours. Thanks to the reviewer’s useful comments and suggestions, we believe that our manuscript definitely improved considerably. Please do not hesitate to let us know if there is something we should correct or reinforce.

Reviewer 2 Report

Line 106, 221, 294 and 358 should conclude with a full-stop.

Line 162 should clarify what you mean by "semiconduct operations".

I think line 241-8 should be moved to literature review/motivation.

Line 293-4 should be rewritten as ""...or other activities of life cannot be assessed or are not considered for this compensation program."

Line 348-50. Consistent with before, should you use plural nouns?

Author Response

Reviewer 2

The author expressed special thanks to you for 2nd useful comments and suggestion, improving definitely the quality of our manuscript

1

Line 106, 221, 294 and 358 should conclude with a full-stop.

Reply

We have corrected all of these.

2

Line 162 should clarify what you mean by "semiconduct operations".

Reply

Typo. We changed it into semiconductor operations

3

I think line 241-8 should be moved to literature review/motivation.

Reply

In accordance with your suggestions, these sentences were moved to introduction part (please see lines 51-56 on page 2)

Line 293-4 should be rewritten as ""...or other activities of life cannot be assessed or are not considered for this compensation program."

Reply

In accordance with your suggestions, we have changed it as you suggested (please see the line 283-284 on page 11)

Line 348-50. Consistent with before, should you use plural nouns?

Reply

We checked this sentence. We corrected welfare programs in line 351 on page 12 to welfare program. In accordance with your comments, we are going to review the entire part of the manuscript and maintain the consistency in use of several key words including the compensation program, WRD etc. in the next step.